# Profit-Influencing Factors in Orthopedic Surgery: An Analysis of Costs and Reimbursements

**DOI:** 10.3390/ijerph19074325

**Published:** 2022-04-04

**Authors:** Felix Rohrer, Aresh Farokhnia, Hubert Nötzli, Frederik Haubitz, Tanja Hermann, Brigitta Gahl, Andreas Limacher, Jan Brügger

**Affiliations:** 1Centre Hospitalier Universitaire Vaudois, CHUV, 1011 Lausanne, Switzerland; 2Department of Internal Medicine, Sonnenhofspital, 3006 Bern, Switzerland; janbruegger@sonnenhof.ch; 3Clinic for Immunology, University Hospital Zurich, 8091 Zurich, Switzerland; aresh.farokhnia@gmail.com; 4Orthopädie Sonnenhof, 3006 Bern, Switzerland; hubertnoetzli@sonnenhof.ch; 5Faculty of Medicine, University of Bern, 3012 Bern, Switzerland; 6PricewaterhouseCoopers AG, 8050 Zurich, Switzerland; frederik.haubitz@pwc.ch; 7Stiftung Lindenhof, Campus SLB, Swiss Institute for Translational and Entrepreneurial Medicine, 3010 Bern, Switzerland; tanja.hermann@lindenhofgruppe.ch; 8Clinical Trial Unit, University of Bern, 3012 Bern, Switzerland; brigitta.gahl@ctu.unibe.ch (B.G.); andreas.limacher@ctu.unibe.ch (A.L.); 9Faculty of Medicine, University of Zurich, 8006 Zurich, Switzerland

**Keywords:** Swiss DRG, finance, orthopedic surgery, cost-benefit profitability analysis, patient health data, costing, profit, net financial result

## Abstract

The aging population and the associated demand for orthopedic surgeries are increasing health costs. Although the Diagnostic Related Groups (DRG) system was introduced to offer incentives for hospitals, concerns remain that reimbursements for older and frail patients do not cover all hospital expenses. We investigated further: (1) Does age influence net financial results in orthopedic surgery? (2) Are there patient or surgical factors that influence results? This retrospective, monocentric study compares costs and reimbursements for orthopedic patients in a tertiary care hospital in Switzerland between 2015 and 2017. The data of 1230 patients were analyzed. Overall, the net results for the hospital were positive, despite 19.5% of patients being treated at a loss. We did not find any correlation between age and profitability (*p* = 0.61). Patient-related factors associated with financial losses were female sex (*p* < 0.001) and diabetes (*p* = 0.013). Patients free of serious comorbidities (*p* = 0.012) or with a higher cost weight (*p* < 0.001) were more often profitable. A longer length of stay was associated with higher losses (*p* < 0.001). This is the first study to address the Swiss DRG reimbursement system in a broad orthopedic population, while also analyzing specific patient and surgical factors. Overall, the reimbursement system is fair, but could better account for certain interventions.

## 1. Introduction

The Federal Statistical Office of Switzerland predicts that the number of people between 65 and 79 years will increase by 53% by 2060. Individuals over 80 will rise even more, making up 12% of the Swiss population by 2060 [1]. An aging population results in an increased demand for orthopedic interventions. Indeed, Kurtz et al. predicted the demand for total knee and hip arthroplasties (TKA/THA) in the US will rise significantly between 2005 and 2030. Primary THA was projected to increase by 174% to 572,000. If TKA numbers continue to increase at current rates, demand is projected to grow by 673% to 3.48 million procedures [2]. Already in 1997 costs due to hip fractures in the US were estimated at a staggering 20 billion dollars [3]. As the population ages, these numbers will become even more significant. To contain the increased health care costs due to the emergence of multimorbid and elderly surgical patients, several reimbursement models have been developed to both ensure qualitatively high and adapted medical care and to control costs and increase efficiency. One of them is bundle payments, divided into Diagnostic Related Groups (DRG). This reimbursement model is used in Switzerland (as in most European countries and in the US). DRG is a patient classification system that defines, amongst other things, the cause of hospitalization, procedures performed, lengths of stay, and levels of complexity (e.g., multimorbidity as a secondary diagnosis). For a THA for example, DRG determines the average financial “cost weight” or from a medical point of view, the Case Mix (CM). The cost weight is then multiplied by the base rate, which due to negotiations can differ slightly depending on the hospital and health insurer. This, combined with government guidelines results in an effective reimbursement rate for the hospital. The cost weight is adapted according to the severity of the disease and length of stay (LOS). For example, if patients stay longer or shorter than the DRG projected LOS range, which is based on average patients in this group, they are called outliers and the cost weight is adapted. Thus, if the patient can be discharged early within the defined range, the hospital performs well and earnings are potentially higher. If the patient leaves before or after the defined range, the cost weight is adapted.

DRG-based reimbursement was introduced in Switzerland in 2012 and debated intensively. The main discussion point is that patients with multiple co-morbidities use more resources and have a longer LOS [4]. Thus, the crucial question for a hospital is: Are costs for older and frail patients sufficiently covered by reimbursements under the DRG system? In the US, a recent analysis comparing revision total joint arthroplasties (TJA) with primary TJA concluded that comorbidities or complications in revision TJA were not fully compensated by the current reimbursement model [5]. Additionally, a retrospective review of 1800 TJA concluded that the bundled payment model did not take frail and older patients into account enough–possibly discouraging care for this high-risk group as they generate more costs [6]. However, solid evidence is not available to confirm such a conclusion, especially in a broader population. Moreover, data on hospital revenues for elderly and frail orthopedic patients treated under the Swiss DRG system-originally based on the German model is lacking.

Therefore, we investigated the reimbursements of costs for older and high-risk patients under the Swiss DRG model and asked: (1) Does age influence profitability in orthopedic surgery? and (2) Are there patient or surgically related factors that influence the net financial result?

## 2. Methods

### 2.1. Study Design

We compared hospital costs to DRG reimbursements for orthopedic patients in a tertiary care hospital between November 2015 and September 2017. The primary outcome was the net financial result. We focused on the effect of age and also identified patient and surgical factors or cost points that influenced the bottom line. In a nutshell, “profitable” patients were compared to “unprofitable” ones. The study protocol was approved by the local Ethics Committee (PB_2016_00256).

### 2.2. Participants

Patients included in this study were initially recruited for the DECO-SSI (DECOlonisation and surgical site infections) randomized controlled trial (RCT) which investigated the impact of preoperative decolonization on the occurrence of SSI [7]. All patients undergoing an orthopedic intervention were screened for study participation. Inclusion criteria were: a minimum age of 16 years and a period of at least 14 days before surgery to enable preoperative screening for S. aureus colonization. Exclusion criteria were: allergy to mupirocin or chlorhexidine, the presence of a foreign nasal body, pregnancy, or planned intervention for a documented infection. Of the 1318 patients included in the RCT, we further excluded outpatient interventions (not reimbursed by bundled payments), patients with missing administrative or billing information, and/or incomplete health data. After exclusion, 1230 cases with complete data sets remained.

### 2.3. Detailed Study Protocol

In this retrospective and observational study, we analyzed three main sources: patients’ health data, treatment costs, and administrative data. All relevant information was entered into the secure web data storing system REDCap (Research Electronic Data Capture, Version 8.5.19, Vanderbilt University, Nashville, TN, USA).

Health data originated from the initial DECO-SSI trial. Patient characteristics were surveyed prospectively. Surgical characteristics were retrospectively extracted from the electronic patient file system (KISIM, Cistec AG, Zurich, Switzerland).

Patients were classed according to age groups below 50 years, between 50–59 years, 60–69 years, 70–79 years, and older than 80 years. We made this decision in accordance with the Age-adjusted Charlson Comorbidity Index (AACCI), which adds a point for every age group above 50 years.

Costs were provided by the medical controlling and billing department. They consisted of 35 cost units. We listed the most important positions of interest separately. They included: costs of medication, implants and materials, personnel, and laboratory and diagnostic tests. The remaining expenses titled “Other Costs” included blood products, patient transportation by third parties, patient administration, operating room, anesthesia, ICU, IMC and emergency room, dialysis, expenses for lodging (room keeping, kitchen, etc.), nutrition counseling, and physiotherapy.

The net financial result was defined as the difference between hospital output/expenses and reimbursement. Patients were then divided into “profitable” or “unprofitable” according to the net financial result for the hospital.

The administrative data included patient LOS, the DRG projected LOS and the cost weight.

### 2.4. Statistical Analysis

We used univariable and multivariable linear regression to investigate the association of patient age and comorbidities with net financial results. The variables selected for multivariable analysis were made for conceptual reasons and were not data driven. Our question was whether older patients might incur costs that are not reimbursed by insurance companies. Age and cost weight were included in the model for this reason. We also added the variable “healthy” as it affects treatment costs that may not be covered by cost weight. Continuous variables are shown as median with lower and upper quartile, and comparisons were made using the Mann–Whitney test. Categorical data are shown as number/% and compared using Fisher’s exact test. All analyses were carried out using Stata 16 (Stata Corp., College Station, TX, USA). A *p*-value of <0.05 was regarded as significant.

## 3. Results

Demographic data of the 1230 included patients is provided in Table 1. One strongly deficitary outlier case (CHF 12,000 loss due to complications) was excluded from parametric analyses and all figures. Each age group contained between 21–31% of patients, except those over 80 accounted for only 2.6% (32 of 1230 patients). Overall, 240 of 1230 (19.5%) surgeries were unprofitable and 990 made earnings for the hospital. There were 10 readmissions defined by a rehospitalization within 18 days after hospital discharge-seven of which were deficitary.

### 3.1. Primary Aim

There was no significant association between net financial results and age (*p* = 0.61) (Table 2 and Figure 1). In hip surgery, however, earnings decreased by CHF 22 per year of patient age (adjusted for connective tissue disease, no comorbidities and cost weight) (Table 3), but the interpretation of such subgroup analysis should be made with caution. Older age increased the total cost of stay (Figure 2), partly due to higher nursing costs (Figure 3) and longer LOS (Figure 4). As expected, older age was associated with a higher cost weight (Figure 5) and a higher AACCI (Figure 6), representing the more multimorbid older population. Similarly, costs and AACCI scores were also correlated (Figure 7).

### 3.2. Secondary Aim

Patient factors associated with unprofitable outcomes were: (1) Female sex with 63% (151 of 240 unprofitable cases) vs. 50% males (492 of 990 profitable cases), *p* < 0.001, and (2) Diabetes with 10% (25 of 240 unprofitable cases) vs. 5.7% (56 of 990 profitable cases), *p* = 0.013. (Table 1). Patients categorized as “healthy”-defined by the absence of diabetes, heart disease, chronic obstructive pulmonary disease (COPD), cerebrovascular, or liver disease-were more often profitable with 85% (840 of 990 patients) vs. 78% (187 of 240 patients), *p* = 0.012 (Table 1). This classification into “healthy” was chosen to better account for major current comorbidities. Nevertheless, no association could be found between the AACCI score and net financial result (median AACCI score 2 in profitable and unprofitable cases) (Table 2 and Figure 8).

Orthopedic sub-specialties and their association with net financial results are indicated in Table 1. Spine surgery, shoulder arthroplasty, knee revision and foot surgery, in particular, were associated with negative net financial results. Moreover, the cost weight in spine surgery showed both higher median and variability than in other surgery types, as illustrated in Figure 9. Surgical factors associated with profitability were primary hip and knee surgery (incl. primary TKA and knee arthroscopy). The distribution of net financial results in relation to the surgical site is detailed in Figure 10.

Univariable and multivariable analyses are shown in Table 3. Considering the entire cohort, age and “healthy” patients (i.e., with no major comorbidities) were not associated with net financial results. Revenue increased on average by CHF 1026 per cost weight. An increase in revenue per cost weight was even more evident in shoulder and knee surgery but was not found in other surgery sites. Including all variables in the model found “healthy” patients significantly associated with positive net financial results. With respect to spine surgery and foot surgery, none of these variables showed any association with net financial results. In patients undergoing hip surgery, patient age was associated with decreased revenue of CHF 22 per patient year, whereas no comorbidities and cost weight were associated with an increase by CHF 923 and CHF 2256, respectively. Only cost weight showed an association in shoulder and knee surgery.

Table 4 shows cost units and their distribution between the “no deficit” and “cost deficit” groups. Associated with unprofitable cases were notably higher nursing and peri-interventional costs. In contrast, cost points associated with profitable cases were a higher doctor’s fee. Figure 11 provides an overview of relative cost distributions. Major cost points are implants, personnel costs and doctor’s fees. Variation occurs between prosthetic surgery and orthopedic sub-specialties using fewer implants.

Furthermore, a higher cost weight (1.7 [1.2–2.0] vs. 1.0 [0.63–2.0]) was observed in profitable cases, *p* < 0.001. LOS was longer in unprofitable patients (5 days [4–7] vs. 4 days [3–5], *p* < 0.001) (Table 5). The DRG projected LOS was also lower in unprofitable patients (5.7 days [2.9–7.9] vs. 7.6 days [6.0–7.9]. In summary, unprofitable patients had an actual LOS approaching the DRG projected LOS (5 vs 5.7 days). The relation between costs and reimbursements for the various surgery types is shown in Figure 12. In general, costs and reimbursement are proportional and the hospital is able to make some profit (as indicated by the lines on the “reimbursement” side of the dashed line (right side). The illustration also indicates that spine surgery has less margin in general.

## 4. Discussion

### 4.1. Primary Aim

Age was not found to be a predictor for unprofitable orthopedic interventions in this retrospective analysis. To our knowledge, this is the first study to directly compare net financial results in different age groups in a broad orthopedic population. Contrary to the hypothesis that higher costs in older patients lead to insufficient net income, this was not confirmed in our study. In this regard, the Swiss DRG payment model appears to be appropriate and well balanced.

### 4.2. Secondary Aim

Patient and surgically related factors, as well as various cost positions, were analyzed. By comparing costs and reimbursements in relation to health data, we were able to identify several factors associated with profit or loss for the hospital. This study provides a detailed account of net financial results in orthopedic surgery under the Swiss DRG. Comparison with existing literature is difficult, as there is a lack of data for the Swiss or German DRG system. As the principles are the same, we compared our data with other bundle payment models from Australia, Canada, Malaysia and the USA.

Patients without major comorbidities, classified here as “healthy”, were more profitable for the hospital than those with pre-existing conditions. This could incentivize hospitals or physicians to (pre)select specific cases for elective orthopedic surgery. Likewise, patients with higher cost weights (i.e., with a higher burden of treatment and costs) also frequently generated earnings for the hospital. This could, therefore, be seen as an incentive to treat less healthy patients. The question, therefore, arises: Why did loss-making patients have a lower cost-weight? A possible explanation could be incomplete documentation, leading to a lower coded costs-weight despite a high treatment burden. It is, therefore, important that complete medical documentation and coding are ensured. This was demonstrated in Internal Medicine in the Australian DRG system, where a review of clinical documentation in 150 cases resulted in a revision of DRG and cost weight-with potential gains of 142,000 AUD [8]. Insufficient clinical coding can lead to losses, as shown in a Malaysian DRG system analysis, which found coding errors in 415 of 424 patients, resulting in inaccurate DRG codes in 74% of cases [9]. Overall, our study found no incentive to treat only “healthy” or only “unhealthy” patients, as both can be valuable for a hospital. This is underlined by the finding that median net financial results do not increase with a higher ACCI score (Figure 7), despite higher total costs per ACCI score (Figure 8).

With regards to specific interventions, a strength of this study is the inclusion of a broad range of orthopedic interventions. We showed that overall, primary hip and knee surgery were profitable, whereas spine surgery and revision knee were more often performed at a loss. Spine surgery in particular shows higher fluctuations in profit/loss than other interventions (Figure 9), as well as a higher cost weight with greater variability (Figure 10). Indeed, Hines et al.’s narrative review categorize spine patients as very heterogeneous with regard to surgery and postoperative outcomes [10]. They concluded that bundle payment systems could produce benefits in this surgery subtype, but should be stratified for risks and treatment costs. According to our results, the Swiss DRG does not account enough for this variability, resulting in significant fluctuations in net financial results. Ideally, cost weight should adapt reimbursement to match actual costs, and not have the benefit making patients subsidize the loss making patients. Furthermore, in knee surgery, we found clear differences between primary knee intervention (more profitable) and revision surgery (less profitable). Similar results were found by Fang et al., which compared the incomes for TJA and revision TJA in 13,000 cases [5]. Hospital costs for revision surgery were proportionally higher than the reimbursements. Cost deficient surgeries can create negative incentives and could potentially limit access to this kind of surgery for patients in need. This is of special importance considering the increasing numbers of TJA-and revision TJA. Our data shows that the current reimbursement system encourages such kind of incentives to specialize in more profitable interventions.

Independent of the orthopedic sub-specialty, the LOS strongly influences the net financial results of a hospital. Globally, 80% of surgeons indicated they were under pressure to decrease costs of THA and 68% encouraged to reduce LOS [11]. Our study shows more specifically that the actual LOS of patients associated with a deficit approaches the DRG projected LOS, leaving less or no possibility for net financial gains. This is a limitation of the DRG system as if all hospitals perform well, DRG projected length of stay is shortened for specific interventions. This results in less financial gain and additional pressure to shorten actual LOS again. Some hospitals will be unable to follow the trend and suffer losses, forcing them to abandon certain interventions, which may result in less health care accessibility for the general population.

If LOS cannot be reduced to optimize financial results, other major cost points must be reduced. In our study, the main cost positions were implants, personnel, and “other costs”, with some variability between surgical subspecialties, as shown in Figure 11. Indeed, a survey estimating the costs for a TKA in Austria, Germany, and Switzerland found that personnel made up the majority of all costs in Switzerland [12]. A logical consequence for hospitals is to reduce personnel costs and increase pressure on medical staff to be efficient-resulting in less time to spend with the patient and diminished human interactions. The effect of this pressure on chronically understaffed hospitals was dramatically shown during the Covid-19 pandemic.

An alternative way to cut costs was suggested by a study assessing primary THA in nine EU countries. They concluded that the main cost drivers are implants (34%), ward costs (20.9%) and cost of surgery (12.9%) [13]. Indeed, when surgeons are asked ways to reduce costs, 30% stated that negotiating a reduced implant price from the supplier was the most important measure [11]. Negotiating reduced implant prices is a possible cost reduction point, as implants make up roughly one third of the costs in spine, knee and hip surgery. This is one possibility applicable for prosthetic surgery, without diminishing the quality of patient care.

### 4.3. Limitations

The main limitation of this study is the single center design. As net financial results depend on hospital performances as well as cost allocation design, this may differ between hospitals. The hospital in question was awarded the Swiss hospital certification of business accounting called REKOLE^®^, which includes cost calculation. This standard indicates that the present hospital has comparable cost allocation to other REKOLE^®^-certified hospitals-which was also confirmed by the coherence of our data with other findings.

Another limitation is the retrospective design of this study. Patient health information, however, was collected prospectively and the hospital controlling department provided other data, both sources ensuring real life conditions in a DRG system. Furthermore, patients included in this analysis were recruited initially for an RCT, which could theoretically introduce a selection bias, as patients included in trials may be healthier than the general population. To account for this the variable “healthy” was added to the multivariate analyses.

Doctors’ fees represent the cost of the treating physician. However, according to the Swiss remuneration model, reimbursements for physicians depend on the insurance class of the patient. In basic health insurance, physician reimbursements are covered by the DRG. For privately insured patients’ additional fees not included in the DRG can be charged. Thus, theoretically, this could be an incentive to prolong LOS. In this study we did not distinguish the patients’ class of insurance, therefore, we cannot directly account for this bias. Nevertheless, in the hospital where the study was conducted, the bed occupancy is very high and hospitalizing patients longer than medically indicated means that no new admissions are possible. Finally, medical reasons define LOS and financial incentives most likely do not encourage longer LOS. The results confirm this, as the actual median LOS was lower than the DRG projected LOS.

One could argue that hospitals should increase the efficiency of all procedures producing losses, but our study shows that variabilities between different procedures (despite the same care management) indicate that there are most likely factors in the reimbursements models not adequately accounted for. In addition, it is not always possible to predict which treatment will be necessary ahead of time-and consequently difficult to anticipate the earning potential.

A more general limitation of this study is the singular focus on treatment profitability-and not on the hospital as a whole. Chen et al. noted that providing unprofitable services could fulfill various other business needs, such as coverage of cost items (infrastructure, staff, etc.) and reserved capacities, preserving a good reputation, or/and the ability to market offerings from a wide range of services [14]. In turn, the hospital becomes all the more attractive for “profitable” treatments. Furthering education and workplace attractiveness could also be considered as creating a better environment when skilled workers are in demand. Simply dividing treatments into profitable and unprofitable becomes even more complex when considering that hospitals are organizations that must make earnings to finance investments in the future as well.

## 5. Conclusions

In this large analysis of different orthopedic procedures, no association between age and net financial results could be found. Overall, the Swiss-DRG reimbursement system is balanced according to our data. Nevertheless, there are patient and surgical factors associated with deficient cases. Especially in spine surgery, the reimbursement system does not seem to account for the high variability in patients and surgical treatments. Furthermore, primary surgery is more beneficial than revision surgery, which may lead to incentives not to treat complications. It is of utmost importance that health care regulators adapt DRG reimbursements carefully to leave some scope for hospitals, in order to ensure future investments and ongoing high quality patient care.

## Figures and Tables

**Figure 1 ijerph-19-04325-f001:**
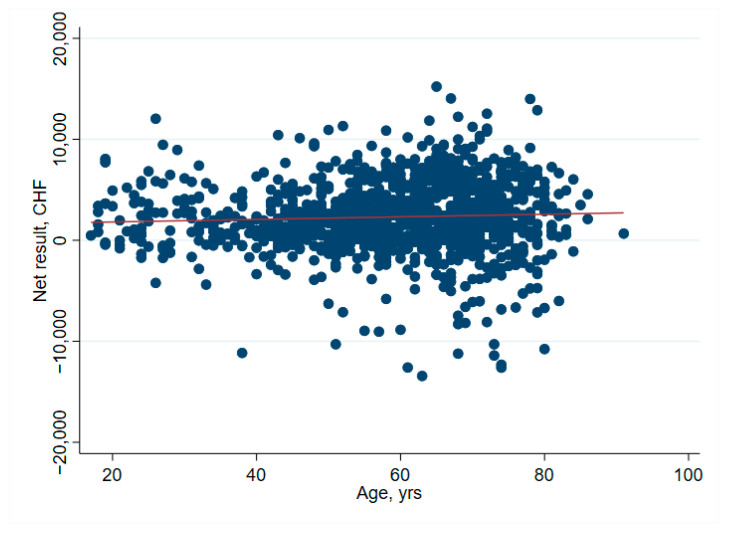
Correlation between age and net financial results. The red line indicates the fitted values.

**Figure 2 ijerph-19-04325-f002:**
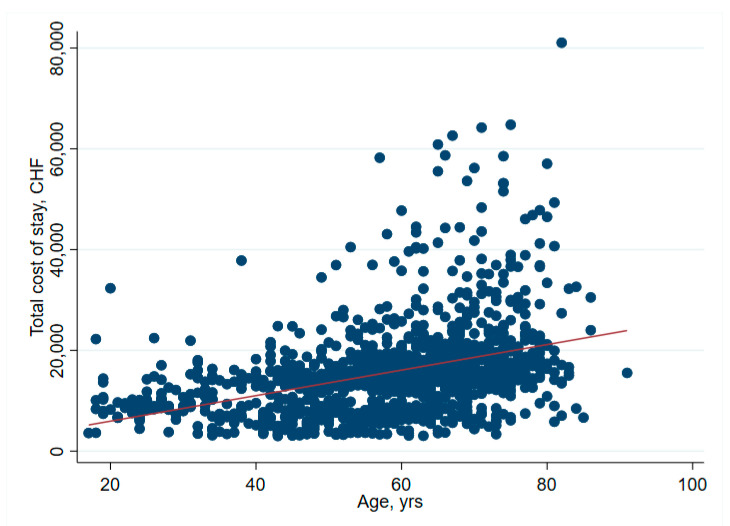
Correlation between age and total cost of stay-total cost of stay increased proportionally with age.

**Figure 3 ijerph-19-04325-f003:**
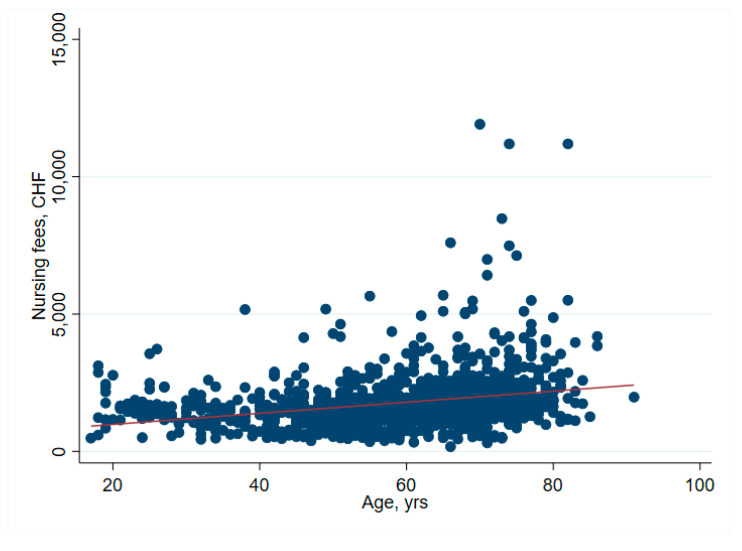
Correlation of age and nursing costs-nursing costs increase with age.

**Figure 4 ijerph-19-04325-f004:**
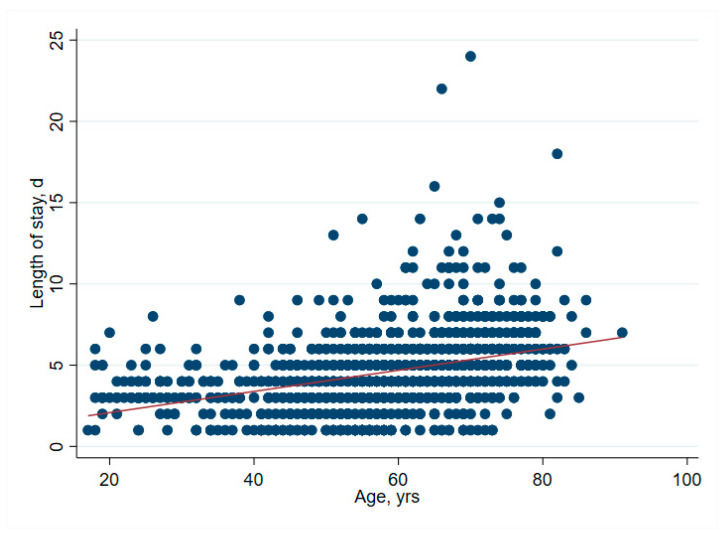
Correlation between age and LOS-LOS increases proportionally with age.

**Figure 5 ijerph-19-04325-f005:**
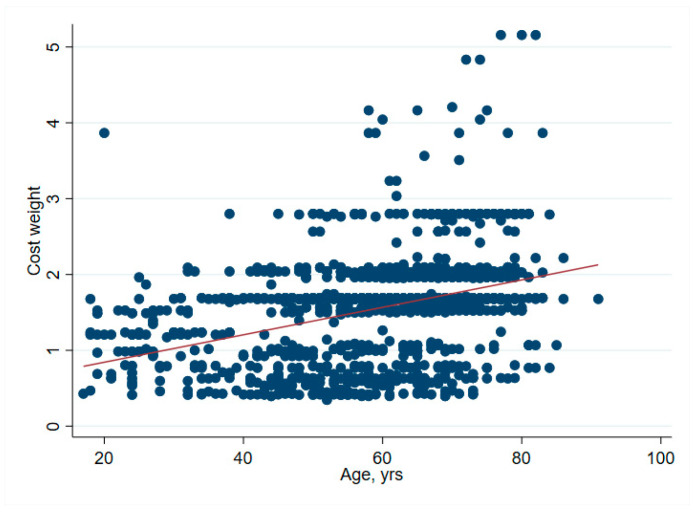
Correlation between age and cost weight. Cost weight increases proportionally with age, indicating the increased burden of treatment and comorbidities associated with older patients.

**Figure 6 ijerph-19-04325-f006:**
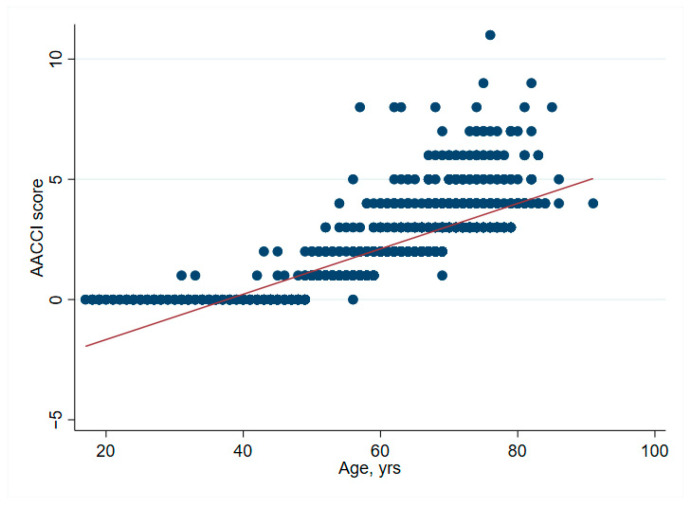
Correlation between age and the age-adjusted Charlston Comorbidity Index. This correlation is logical, as the score adds one point per 10 years after 50 years.

**Figure 7 ijerph-19-04325-f007:**
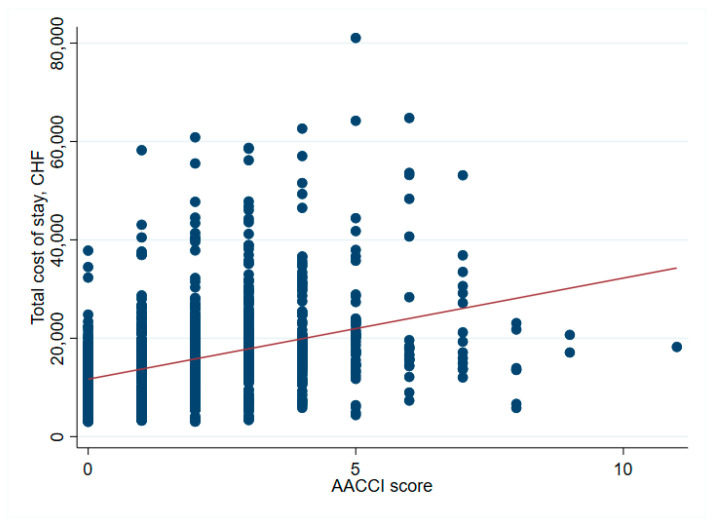
Correlation between total cost of stay and AACCI-cost of stay increases with more comorbidities.

**Figure 8 ijerph-19-04325-f008:**
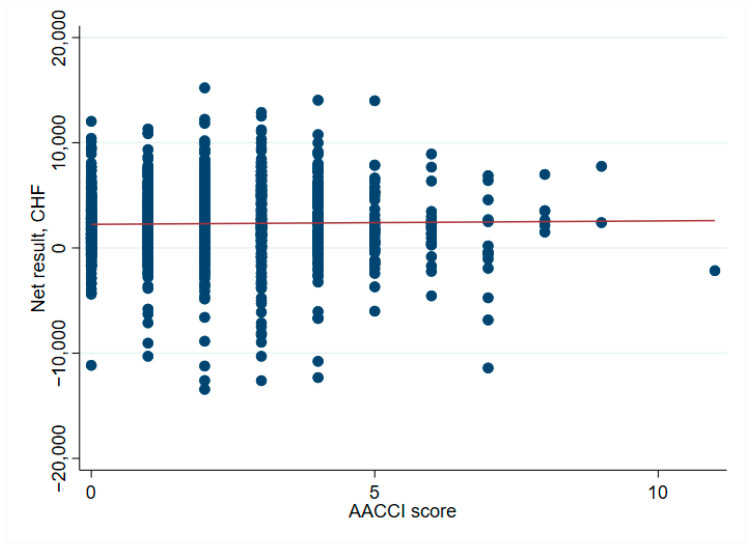
AACCI and net financial results-no statical relevant correlation.

**Figure 9 ijerph-19-04325-f009:**
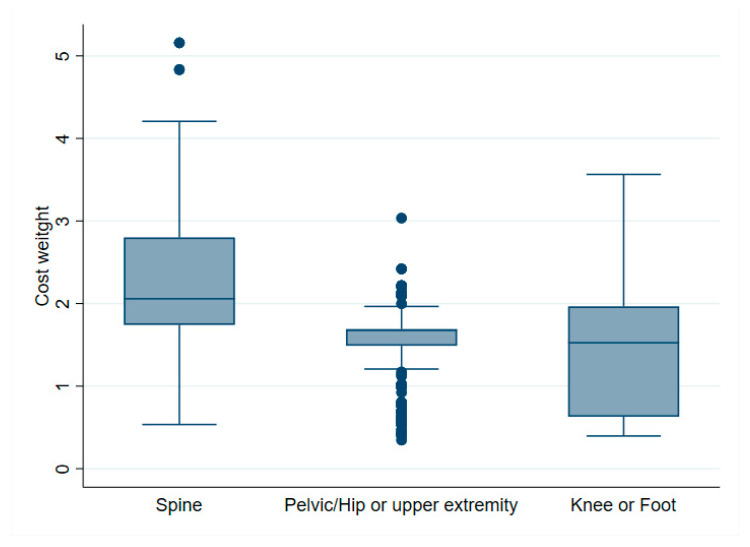
Box plot of cost weight according to surgical site.

**Figure 10 ijerph-19-04325-f010:**
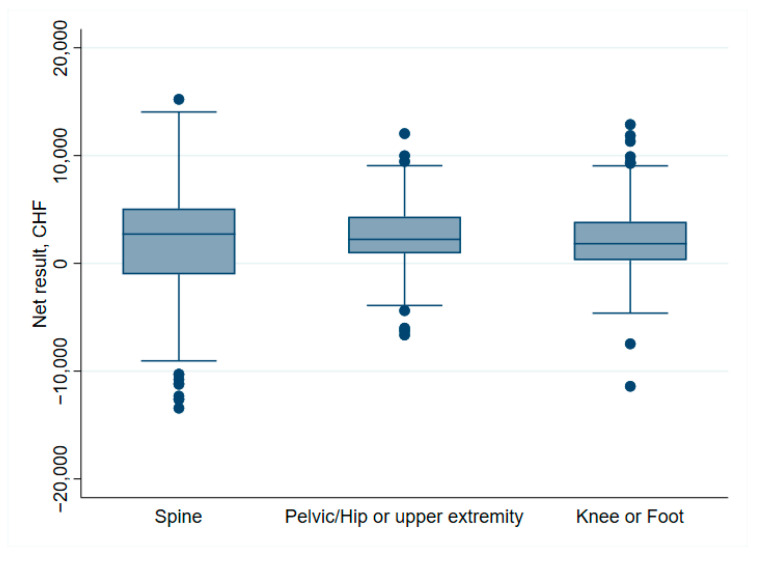
Box plot of net financial results according to surgical site.

**Figure 11 ijerph-19-04325-f011:**
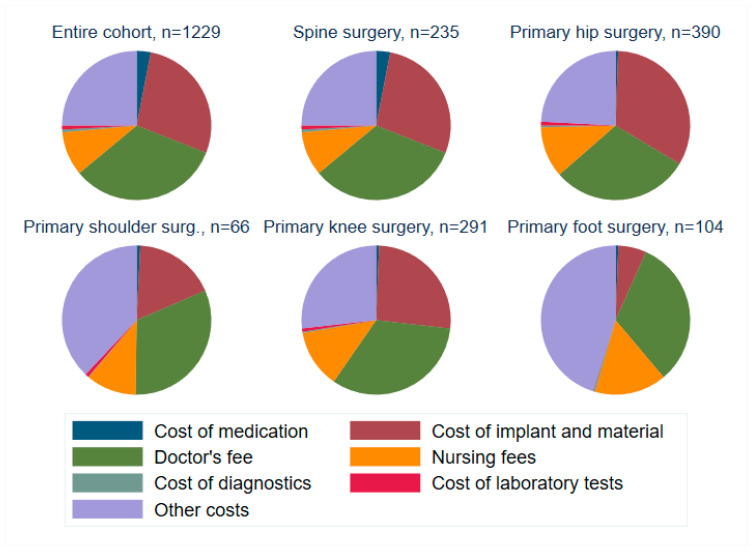
Relative cost distribution by cost units for main types of surgery.

**Figure 12 ijerph-19-04325-f012:**
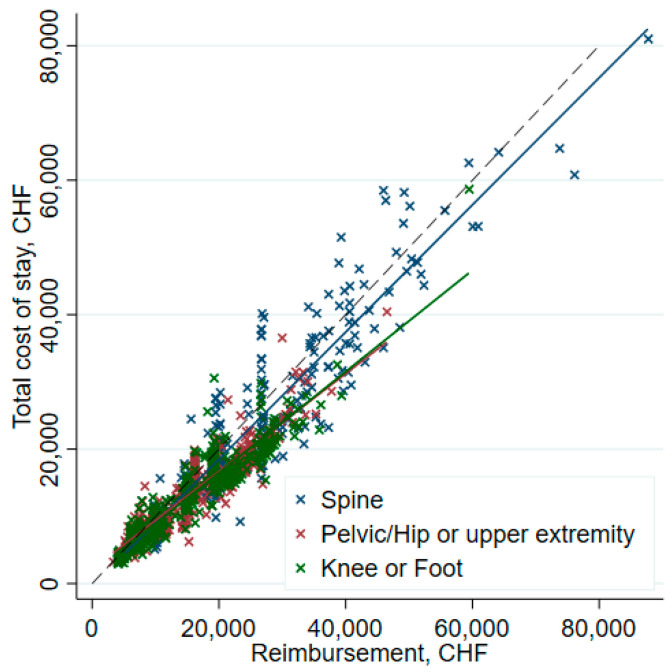
Costs and reimbursement by type of surgery. Dashed line indicates net financial zero. Dots to the right of the dashed line indicate a positive net financial result. Lines in color are regression lines by surgery site.

**Table 1 ijerph-19-04325-t001:** Patient characteristics and surgical sub-types. Patients were divided into two groups: If net financial results were positive, the patient was allocated to the “no deficit” group. If net financial results were negative, the patient was allocated to the “cost deficit” group. *p*-value refers to the difference between the “no deficit” and the “cost deficit” group.

	*n*	Total (*n* = 1230)	No Deficit (*n* = 990)	Cost Deficit (*n* = 240)	*p*
Female sex	1230	649 (53%)	498 (50%)	151 (63%)	<0.001
Active smoker	1230	208 (17%)	165 (17%)	43 (18%)	0.63
Regular alcohol intake	1230	380 (31%)	313 (32%)	67 (28%)	0.28
Alcohol amount (>2 units/day)	380 *	33 (2.7%)	27 (2.7%)	6 (2.5%)	1.00
BMI (kg/m^2^)	1230	26 [24,30]	26 [24,30]	27 [24,30]	0.91
Renal insufficiency	1230	11 (0.89%)	8 (0.81%)	3 (1.3%)	0.46
Diabetes	1230	81 (6.6%)	56 (5.7%)	25 (10%)	0.013
Healthy	1230	1027 (83%)	840 (85%)	187 (78%)	0.012
Type of procedure	1230				<0.001
Spine surgery		236 (19%)	162 (16%)	74 (31%)	
Pelvic/Hip or upper extremity		516 (42%)	449 (45%)	67 (28%)	
Knee or Foot		478 (39%)	379 (38%)	99 (41%)	
Spine surgery	236 **				0.015
Decompression		41 (3.3%)	34 (3.4%)	7 (2.9%)	
Stabilisation		93 (7.6%)	55 (5.6%)	38 (16%)	
Decompression & Stabilisation		91 (7.4%)	63 (6.4%)	28 (12%)	
Other		11 (0.89%)	10 (1.0%)	1 (0.42%)	
Hip or shoulder surgery	516 **				<0.001
Hip: Primary		390 (32%)	368 (37%)	22 (9.2%)	
Hip: Revision		28 (2.3%)	22 (2.2%)	6 (2.5%)	
Hip: Other		3 (0.24%)	3 (0.30%)	0 (0.00%)	
Shoulder: Primary		66 (5.4%)	36 (3.6%)	30 (13%)	
Shoulder: Revision		11 (0.89%)	6 (0.61%)	5 (2.1%)	
Shoulder: Other		11 (0.89%)	10 (1.0%)	1 (0.42%)	
Upper Extremity: Other		7 (0.57%)	4 (0.40%)	3 (1.3%)	
Knee or Foot surgery	478 **				<0.001
Knee: Primary		291 (24%)	268 (27%)	23 (10%)	
Knee: Revision		50 (4.1%)	37 (3.7%)	13 (5.4%)	
Knee: Other		11 (0.89%)	9 (0.91%)	2 (0.83%)	
Foot: Primary		104 (8.5%)	50 (5.1%)	54 (22%)	
Foot: Revision		12 (0.98%)	7 (0.71%)	5 (2.1%)	
Foot: Other		10 (0.81%)	8 (0.81%)	2 (0.83%)	

* In this row, 380 of 1230 patients indicated regular alcohol consumption. Analysis aimed to elucidate if higher alcohol consumption (i.e., more than 2 units of alcohol/day) impacted net financial results. ** Number of patients in this orthopedic sub-speciality.

**Table 2 ijerph-19-04325-t002:** Age-adjusted Charlston Comorbidity Index components. AACCI components are listed below. *p*-value refers to the difference on net financial results between both groups.

	Total (*n* = 1230)	No Deficit (*n* = 990)	Cost Deficit (*n* = 240)	*p*
Age				0.61 *
<50 years	274 (22%)	220 (22%)	54 (22%)	
50–59 years	287 (23%)	237 (24%)	50 (21%)	
60–69 years	383 (31%)	311 (31%)	72 (30%)	
70–79 years	254 (21%)	196 (20%)	58 (24%)	
>80 years	32 (2.6%)	26 (2.6%)	6 (2.5%)	
Myocardial infarction	49 (4.0%)	37 (3.7%)	12 (5.0%)	0.36
Congestive heart failure	39 (3.2%)	28 (2.8%)	11 (4.6%)	0.21
Peripheral vascular disease	22 (1.8%)	14 (1.4%)	8 (3.3%)	0.06
CVI or TIA	47 (3.8%)	36 (3.6%)	11 (4.6%)	0.46
Dementia	0	0	0	
COPD	20 (1.6%)	16 (1.6%)	4 (1.7%)	1.00
Connective tissue disease	57 (4.6%)	44 (4.4%)	13 (5.4%)	0.50
Peptic ulcer disease	7 (0.57%)	3 (0.30%)	4 (1.7%)	0.030
Liver disease				0.58 *
none	1226 (100%)	987 (100%)	239 (100%)	
mild	3 (0.24%)	2 (0.20%)	1 (0.42%)	
moderate to severe	1 (0.08%)	1 (0.10%)	0 (0.00%)	
Diabetes mellitus				0.07 *
none or diet-controlled	1151 (94%)	934 (94%)	217 (90%)	
uncomplicated	67 (5.4%)	47 (4.7%)	20 (8.3%)	
end-organ disease	12 (0.98%)	9 (0.91%)	3 (1.3%)	
Hemiplegia	3 (0.24%)	1 (0.10%)	2 (0.83%)	0.10
Moderate to severe CKD	4 (0.33%)	2 (0.20%)	2 (0.83%)	0.17
Solid tumour				0.80 *
none	1162 (94%)	937 (95%)	225 (94%)	
localized	63 (5.1%)	49 (4.9%)	14 (5.8%)	
metastatic	5 (0.41%)	4 (0.40%)	1 (0.42%)	
Leukaemia	8 (0.65%)	7 (0.71%)	1 (0.42%)	1.00
Lymphoma	0	0	0	
AIDS	0	0	0	
AACCI score	2.0 [1.0–3.0]	2.0 [1.0–3.0]	2.0 [1.0–3.0]	0.16

* *p*-value refers to overall calculations for the variable (e.g., age). As the categories of one variable are dependent of each other, no different result is expected for each category. As *n* were small for single categories, we did not compare one single category with the other categories.

**Table 3 ijerph-19-04325-t003:** Univariable and Multivariable analysis. Effect of each variable on net financial results.

	Variable	Difference in Net Financial Result (in CHF)	*p*
Univariable			
Entire cohort, *n* = 1229	Age per year	13 (−1–26)	0.068
Entire cohort, *n* = 1229	Healthy	341 (−176–859)	0.196
Entire cohort, *n* =1 229	Cost weight	1026 (758–1293)	<0.001
Spine surgery, *n* = 235	Cost weight	−56 (−879–766)	0.893
Primary hip surgery, *n* = 390	Cost weight	1017 (−226–2259)	0.108
Primary shoulder surgery, *n* = 66	Cost weight	2030 (934–3127)	<0.001
Primary knee surgery, *n* = 291	Cost weight	1992 (1475–2508)	<0.001
Primary foot surgery, *n* = 104	Cost weight	−1044 (−3549–1462)	0.411
Multivariable			
Entire cohort, *n* = 1229	Age per year	−4 (−19–11)	0.614
	Healthy	544 (24–1064)	0.040
	Cost weight	1082 (796–1369)	<0.001
Spine surgery, *n* = 235	Age per year	−28 (−90–34)	0.372
	Healthy	−640 (−2403–1123)	0.475
	Cost weight	11 (−843–864)	0.980
Primary hip surgery, *n* = 390	Age per year	−22 (−40–−4)	0.018
	Healthy	923 (304–1542)	0.004
	Cost weight	2256 (784–3728)	0.003
Primary shoulder surgery, *n* = 66	Age per year	−11 (−56–34)	0.629
	Healthy	−353 (−1786–1081)	0.624
	Cost weight	2157 (893–3421)	0.001
Primary knee surgery, *n* = 291	Age per year	20 (−9–48)	0.171
	Healthy	727 (−60–1513)	0.070
	Cost weight	1848 (1256–2439)	<0.001
Primary foot surgery, *n* = 104	Age per year	−14 (−38–10)	0.241
	Healthy	461 (−698–1620)	0.432
	Cost weight	−408 (−3058–2243)	0.761

One outlier was excluded from analysis, as explained in the Results section, resulting in *n* = 1299.

**Table 4 ijerph-19-04325-t004:** Overview of cost units (in CHF), including median costs for each unit for no deficit and the cost deficit groups, and relative *p*-values.

Cost Unit	Total (*n* = 1230) Median [Q1–Q3]	No Deficit (*n* = 990) Median [Q1–Q3]	Cost Deficit (*n* = 240) Median [Q1–Q3]	*p*
Cost of medication	69 [52–99]	68 [52–93]	81 [52–149]	<0.001
Cost of implant and material	5096 [812–5730]	5334 [98–5718]	2039 [553–7778]	0.49
Doctor’s fee	3689 [2990–6704]	3873 [3170–6860]	3309 [1510–4696]	<0.001
Nursing costs	1587 [1213–2058]	1552 [1202–1952]	1766 [1288–2740]	<0.001
Cost of diagnostics	33 [21–80]	33 [25–80]	45 [17–80]	0.85
Cost of laboratory tests	107 [78–147]	107 [82–141]	107 [0.00–197]	0.93
Periintervenional costs	3805 [3151–4873]	3617 [3031–4447]	5146 [3967–7193]	<0.001
Total cost of stay	14,676 [9684–18,501]	14,676 [10,943–18,022]	14,921 [8139–23,656]	0.23
Total reimbursement	16,160 [11,375–22,942]	16,186 [14,521–23,188]	13,842 [6703–20,039]	<0.001
Net result	2147 [422–4435]	2741 [1504–5062]	−1240 [−2811–−492]	n/a

**Table 5 ijerph-19-04325-t005:** Overview of DRG relevant data. Length of stay refers to the actual median length of patient stay, while DRG-projected length of stay refers to the median length of stay allocated by the DRG system for specific interventions.

	Total (*n* = 1230) Median [Q1–Q3]	No Deficit (*n* = 990) Median [Q1–Q3]	Cost Deficit (*n* = 240) Median [Q1–Q3]	*p*
Cost weight	1.7 [0.99–2.0]	1.7 [1.2–2.0]	1.0 [0.63–2.0]	<0.001
Length of stay	4.0 [3.0–6.0]	4.0 [3.0–5.0]	5.0 [4.0–7.0]	<0.001
DRG projected length of stay	7.6 [3.7–7.9]	7.6 [6.0–7.9]	5.7 [2.9–7.9]	<0.001

## Data Availability

Not applicable.

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
