# Peer review of "Profit-Influencing Factors in Orthopedic Surgery: An Analysis of Costs and Reimbursements"

_ijerph, 2022, doi:10.3390/ijerph19074325_

Round 1

Reviewer 1 Report

This is a well written and useful study which describes the influence of age mix on hospitals' financial results for elective orthopedic surgery. 

Major comment

The authors need to address the limitation of their recruitment which was used patients included in a randomized controlled trial. While the research question of the trial was unrelated to hip or knee surgery, patients included in trials tend to be healthier, with fewer comorbid conditions, than the general patient population. This is reflected in Table 2.

I the multivariate regression of Table 3, I understand that no other comorbidity than connective tissue disease had any effect on financial result, is that correct? Did you test all the variables with p<0.2? Any effect of BMI?  The Methods section should provide greater detail on the static analyses performed.

Table 4 is difficult to understand because the effects of production costs and DRG weights are not separated. In the column ‘cost deficit’ for example, the LOS is shorter than in the column (no deficit’ but the DRG weight is also lower, which probably explains the deficit. The important clinical question however is whether the cost weight is inappropriate or has been manipulated by the payer to create an incentive to further decrease LOS for example.

In table 4, the first line ‘length of stay ‘ is the median I suppose? The sentence ‘LOS was longer in unprofitable patients (5 days [4, 7] vs. 4 210 days [3, 5], p<0.001)’ is correct for median but the relationship  goes the other way for mean, with a much bigger difference,

I would investigate further the effect of DRG weights on the profitability by including them in the regression. Also the effect of doctors’ fees would require an explanation in the Methods section. How is the doctor’s fee part of the production cost for the hospital? If the fee is given directly to the doctor, it is a covariate but not a production cost. Are the fees proportional to length of stay or unrelated?

Minor comment

I would try to find a synonym for ‘product’ even though it is quite appropriate. Perhaps ‘output’ or ‘admissions’ since these represent patients’ admitted and treated in hospitals

Author Response

We would like to thank the reviewers for their helpful and insightful comments which provided significant motivation to clarify and rework the manuscript.

Please find our responses, including the corresponding lines and sections in the manuscript, below.

Reviewer 1:

Comment 1.1: “The authors need to address the limitation of their recruitment which was used patients included in a randomized controlled trial. While the research question of the trial was unrelated to hip or knee surgery, patients included in trials tend to be healthier, with fewer comorbid conditions, than the general patient population. This is reflected in Table 2.”

Answer 1.1: This possible bias has been added to the Limitations section. (lines 242-244)

Comment 2.1: “I the multivariate regression of Table 3, I understand that no other comorbidity than connective tissue disease had any effect on financial result, is that correct? Did you test all the variables with p<0.2? Any effect of BMI?  The Methods section should provide greater detail on the static analyses performed.”

Answer 2.1: The variables selected for multivariable analysis were made for conceptual reasons and were not data driven. Our question was whether older patients might incur costs that are not reimbursed by insurance companies. Age and cost weight were included in the model for this reason. We also added the variable “healthy” as it affects treatment costs that may not be covered by cost weight. This explanation has been added to the Methods section. Furthermore, connective tissue disease was included, as over 30% of patients with connective tissue disease require joint surgery. * This variable in our multivariate analysis represents a comorbidity that is fairly common for an orthopaedic surgeon. Connective tissue disease was deleted from Table 3, as it added little information and might have confused the reader. (lines 97-100)

*Ref: MacKenzie C.R., Su E.P. (2013) Total Joint Arthroplasty in the Patient with Connective Tissue Disease. In: Mandell B. (eds) Perioperative Management of Patients with Rheumatic Disease. Springer, New York, NY. https://doi.org/10.1007/978-1-4614-2203-7_20

Comment 3.1: “Table 4 is difficult to understand because the effects of production costs and DRG weights are not separated. In the column ‘cost deficit’ for example, the LOS is shorter than in the column (no deficit’ but the DRG weight is also lower, which probably explains the deficit. The important clinical question however is whether the cost weight is inappropriate or has been manipulated by the payer to create an incentive to further decrease LOS for example.”

Answer 3.1: As you suggested, Table 4 has been split (Figure 4a + 4b) to facilitate understanding. Indeed, the discussion of LOS and cost weight is important. To better account for this we have altered the Results (lines 163-169) and updated the Discussion (lines 185-197, 213-219).

Comment 4.1: “In table 4, the first line ‘length of stay ‘ is the median I suppose? The sentence ‘LOS was longer in unprofitable patients (5 days [4, 7] vs. 4 days [3, 5], p<0.001)’ is correct for median but the relationship goes the other way for mean, with a much bigger difference,…”

Answer 4.1: Indeed, the first line is median with interquartile range (described under “statistical analysis”, line 101). However, we are not quite sure what you are referring to with respect to the mean. We did not report mean LOS in any table or text, and the only figure reporting on LOS shows LOS and age (Figure 4), indicating that LOS is longer in older patients.

Comment 5.1: “I would investigate further the effect of DRG weights on the profitability by including them in the regression. Also, the effect of doctors’ fees would require an explanation in the Methods section. How is the doctor’s fee part of the production cost for the hospital? If the fee is given directly to the doctor, it is a covariate but not a production cost. Are the fees proportional to length of stay or unrelated?”

Answer 5.1: Thank you for pointing this out. Indeed, doctors’ fees are generally included in the DRG reimbursement and do not vary for patients with basic health insurance. As a result, there is no incentive for physicians to influence LOS in this regard as their fee is included in the hospital costs. For privately insured patients there is an additional fee proportional to LOS which could theoretically represent an incentive to hospitalize a patient longer. However, in strictly economic terms this also means that no new patient can be admitted, and no new “financial” case opened. As bed occupancy in the hospital where the study was conducted is very high, we do not feel that this small financial incentive influences LOS, but rather that medical conditions determine when a patient is discharged. As we did not assess insurance type, we have added this issue in the Limitations instead of the Methods section (lines 245-252). Furthermore, privately insured individuals account for only around 10% of all patients hospitalized.

Comment 6.1: “I would try to find a synonym for ‘product’ even though it is quite appropriate. Perhaps ‘output’ or ‘admissions’ since these represent patients’ admitted and treated in hospitals”

Answer 6.1:  This point has been reformulated. (line 40)

Reviewer 2 Report

I enjoyed reading the paper, however, certain parts need to be updated. You have an opportunity (expectation?) to develop a theoretical contribution, instead of just reviewing the lit in a superficial way. How do your findings challenge and extend what we’ve previously known about the cost and reimbursement?

Your lit review is also very long, providing lengthy arguments for each point.

There is really no theoretical background in this study. What is being tested here? 

The paper needs a more cohesive framework to organize your quality ideas and clarify your contributions. It remains highly fragmented, even with longer sections and much lengthier paragraphs now –you still are listing sections without connections/transitions.

Please include practical interpretations of your findings. Beyond simply reporting a significant increase, or % more likely, how meaningful is this effect?

Again, what is the marginal contribution of this paper? Rather than explaining what all other researchers do – instead explain your novelty, what is important and new here, and then explain how it builds upon theirs.

The current ordering/framing is odd and diminishes the reader’s expectations for the findings you’re presenting.

Empirical Concerns:

I am confused about the empirical findings. For example in Table 1, is this a multivariate analysis? If so, why number of obs. are different for different group of variables. 1230, 990, 240, 516, 718..?

In Table 2, are there differences within age groups? or within diseases?

Again, Tables are placed, rather than discussed. In some tables, we don't know what the aim is.

Digits are not consistent in tables. 

In Table 3, what is the number of obs and what is the R2? If that's a multivariate test? That also applied to Table 4.

Figures barely gets any explanation or description. 

Author Response

Reviewer 2:

Comment 1.2: “I enjoyed reading the paper, however, certain parts need to be updated. You have an opportunity (expectation?) to develop a theoretical contribution, instead of just reviewing the lit in a superficial way. How do your findings challenge and extend what we’ve previously known about the cost and reimbursement?”

Answer 1.2: We thank the reviewer for his valuable remarks. Indeed, the manuscript needed improvements in readability and clarity. We have adapted the Introduction, Results, Discussion and Conclusion sections accordingly. We hope this will improve the study’s strengths and readability. Please excuse that no lines are noted for every comment, as corrections necessitated major revisions throughout the whole manuscript.

Comment 2.2: “Your lit review is also very long, providing lengthy arguments for each point.”

Answer 2.2: It has been shortened and reorganized, with some sections being deleted completely.

Comment 3.2: “There is really no theoretical background in this study. What is being tested here?” 

Answer 3.2: The information regarding the aim has been reformulated and made more specific. (Lines 51-63)

Comment 4.2: “The paper needs a more cohesive framework to organize your quality ideas and clarify your contributions. It remains highly fragmented, even with longer sections and much lengthier paragraphs now –you still are listing sections without connections/transitions.”

Answer 4.2: Again, thank you for this valuable comment. Parts in the Introduction, Results and Discussion sections have been reorganized, shortened, and interconnected.

Comment 5.2: “Please include practical interpretations of your findings. Beyond simply reporting a significant increase, or % more likely, how meaningful is this effect?”

Answer 5.2: The whole Results section has been adapted to include more specific descriptions of the findings. Also, we emphasized the implications of the effects in the Discussion.

Comment 6.2: “Again, what is the marginal contribution of this paper? Rather than explaining what all other researchers do – instead explain your novelty, what is important and new here, and then explain how it builds upon theirs.”

Answer 6.2: Please see the answers above.

Comment 7.2: “The current ordering/framing is odd and diminishes the reader’s expectations for the findings you’re presenting.”

Answer 7.2: Please see the answers above.

Comment 8.2: “I am confused about the empirical findings. For example, in Table 1, is this a multivariate analysis? If so, why number of obs. are different for different group of variables. 1230, 990, 240, 516, 718..?”

Answer 8.2: Table 1 reports the findings between the “no deficit” and “cost deficit” group. Multivariate analysis is shown in Table 3. Patient numbers differ in function of the “subgroup”, for example all patients which had spine surgery. Explanatory footnotes have been added to the Table.

Comment 9.2: “In Table 2, are there differences within age groups? or within diseases?”

Answer 9.2: As the categories of one variable are dependent of each other, no different result is expected for each category. Furthermore, N were small for single categories, therefore we refrained from comparing one single category to the other categories. This has been added in an explanatory footnote to Table 2.

Comment 10.2: “Again, Tables are placed, rather than discussed. In some tables, we don't know what the aim is.”

Answer 10.2: Thank you for the observation. We have embedded the tables in the text, and added some explanations, for example lines 142-149. Also, the ordering of the tables and figures was adapted slightly.

Comment 11.2: “Digits are not consistent in tables.“

Answer 11.2: In Table 3, one outlier was excluded, leading to N=1229 instead of 1230. This is described in the Results section, line 106/107 and an explanatory footnote has been added to the Table.

Comment 12.2: “In Table 3, what is the number of obs and what is the R2? If that is a multivariate test? That also applied to Table 4.”

Answer 12.2: Table 3 shows univariate and multivariate tests. N is shown in R1, according to the examined population. In R2 are the variables. A title has been added to the Table to specify and facilitate comprehension.

Comment 12.2: “Figures barely gets any explanation or description.“

Answer 12.2: We have added an explanation to every Figure. Moreover, more direct connections and descriptions including implications of the finding have been added to the text, for example lines 120-123.

Again, thank you very much for the helpful input and comments on our manuscript.

We hope you will find the re-submitted paper with the corresponding corrections suitable for publication and look forward to hearing from you.

Round 2

Reviewer 1 Report

the authors have adressed my comments, they only need to carefully check the manuscript for minor punctuation errors, particularly in the revised parts. 

Author Response

Thank you very much for the useful review! We have reviewed the punctuation and found also some other minor errors. See lines: 54,55, 59, 63, 69, 127, 148, 154, 169, 230, 234, 263, 268, 269, 272, 276, 292, 302, 318, 327, 338, 344, 347, 349, 377

Best regards

Reviewer 2 Report

Thank you

Author Response

Thank you for the useful review and the effort!

Best regards